# Semi-Supervised Learning in Medical MRI Segmentation: Brain Tissue with White Matter Hyperintensity Segmentation Using FLAIR MRI

**DOI:** 10.3390/brainsci11060720

**Published:** 2021-05-28

**Authors:** ZunHyan Rieu, JeeYoung Kim, Regina EY Kim, Minho Lee, Min Kyoung Lee, Se Won Oh, Sheng-Min Wang, Nak-Young Kim, Dong Woo Kang, Hyun Kook Lim, Donghyeon Kim

**Affiliations:** 1Research Institute, NEUROPHET Inc., Seoul 06247, Korea; clarence@neurophet.com (Z.R.); reginaeunyoungkim@neurophet.com (R.E.K.); minho.lee@neurophet.com (M.L.); 2Department of Radiology, Eunpyeong St. Mary’s Hospital, College of Medicine, The Catholic University of Korea, Seoul 06247, Korea; jeeyoungkim@catholic.ac.kr (J.K.); oasis1979@gmail.com (S.W.O.); 3Department of Radiology, Yeouido St. Mary’s Hospital, College of Medicine, The Catholic University of Korea, Seoul 06247, Korea; 22000659@cmcnu.or.kr; 4Department of Psychiatry, Yeouido St. Mary’s Hospital, College of Medicine, The Catholic University of Korea, Seoul 06247, Korea; smwang11@naver.com (S.-M.W.); nakyoung17@gmail.com (N.-Y.K.); 5Department of Psychiatry, Seoul St. Mary’s Hospital, College of Medicine, The Catholic University of Korea, Seoul 06591, Korea; kato7@hanmail.net

**Keywords:** segmentation, deep-learning, FLAIR, T1w, white-matter hyperintensity

## Abstract

White-matter hyperintensity (WMH) is a primary biomarker for small-vessel cerebrovascular disease, Alzheimer’s disease (AD), and others. The association of WMH with brain structural changes has also recently been reported. Although fluid-attenuated inversion recovery (FLAIR) magnetic resonance imaging (MRI) provide valuable information about WMH, FLAIR does not provide other normal tissue information. The multi-modal analysis of FLAIR and T1-weighted (T1w) MRI is thus desirable for WMH-related brain aging studies. In clinical settings, however, FLAIR is often the only available modality. In this study, we thus propose a semi-supervised learning method for full brain segmentation using FLAIR. The results of our proposed method were compared with the reference labels, which were obtained by FreeSurfer segmentation on T1w MRI. The relative volume difference between the two sets of results shows that our proposed method has high reliability. We further evaluated our proposed WMH segmentation by comparing the Dice similarity coefficients of the reference and the results of our proposed method. We believe our semi-supervised learning method has a great potential for use for other MRI sequences and will encourage others to perform brain tissue segmentation using MRI modalities other than T1w.

## 1. Introduction

Automated quantitative metrics of structural magnetic resonance imaging (MRI), such as cortical volume or thickness, have commonly been used as objective indicators of neurodegeneration related to aging, stroke, and dementia. Recently, they have been combined with other MRI-based biomarkers such as white-matter hyperintensity (WMH) for investigation in Alzheimer’s disease (AD) or aging research [1,2].

The WMH biomarker indicates bright areas appear in the white matter on T2 fluid-attenuated inversion recovery (FLAIR) sequences. The etiologies of WMH are diverse, and it is considered primarily a marker of small-vessel cerebrovascular disease. WMH represent increased blood–brain barrier permeability, plasma leakage, and the degeneration of axons and myelin [3].

Larger WMH regions are associated with an accelerated cognitive decline and increased risk for AD [4]. Recent studies suggest that WMH plays a role in AD’s clinical symptoms and there is synergistic contribution of both medial temporal lobe atrophy and WMH on cognitive impairment and dementia severity [5]. Patients with mild cognitive impairment or early AD had concurrent WMH, which indicates a more significant cognitive dysfunction than those with a low WMH burden [5]. WMH predicts conversion from mild cognitive impairment to AD [6]. In addition, WMH has been reported to have a relationship with structural changes and cognitive performance, especially with respect to processing speed, even in cognitively unimpaired participants [7].

In clinical practice, WMH burden is usually estimated using a visual scale such as the Fazekas scale, but this cannot be used as an objective indicator without analyzing the volumetric ratio between WMH and white matter (WM). The quantification of WMH is essential for evaluating the association of WMH burden with cognitive dysfunction and longitudinal changes in WMH volume. Hence, a reliable automated method for measuring WMH and cortical volume would be helpful in clinical practice. Recently, it was reported that WMH progression is associated with changes in cortical thinning. Therefore, automatic measurement methods for WMH burden and cortical volume measurement on FLAIR MRI would be clinically valuable for tracing the longitudinal change in patients with cognitive impairment [8].

Thus, brain structural analysis, especially volumetric analysis, combined with WMH could provide more descriptive information to reveal the relationship between cognitive performance and MRI-based biomarkers. There are various brain tissue segmentation tools for three-dimensional (3D) T1-weighted (T1w) MRIs, such as FreeSurfer [9], SPM [10], and FSL [11]. However, brain tissue segmentation tools for other magnetic resonance sequences (such as FLAIR, susceptibility-weighted imaging (SWI), and gradient echo (GRE) sequences) are rarely developed because their aim is not to measure brain volume or analyze brain morphology precisely.

A deep learning-based automatic segmentation algorithm was first used to diagnose brain tumors and showed a high automatic detection rate and high accuracy [12]. However, accuracy and reproducibility of automatic segmentation methods for WMH have not been achieved because the gold standard of manual segmentation is time consuming to create. It is also difficult to obtain intra- and inter-rater reliability. In reality, it is rare for clinicians to obtain both T1w and FLAIR MRIs because of the burden of scanning time. Ultimately, this situation hinders approaches that perform brain tissue segmentation on non-T1w sequences.

In this study, we propose a brain tissue segmentation method that enables us to obtain trainable brain labels on FLAIR MRIs. With given T1w MRI and FLAIR MRI paired datasets, we initially generated the brain labels on the T1w MRIs and aligned the label to FLAIR MRIs using the co-registration method. Then, the label quality was improved using a semi-supervised learning method [13]. Finally, we trained a deep neural network-based brain segmentation model for FLAIR MRI.

## 2. Materials and Methods

### 2.1. Subjects

This study has the following Institutional Review Board (IRB) approval. As shown in Table 1, our dataset was taken from the Catholic Aging Brain Imaging (CABI) database, which holds brain MRI scans of outpatients at the Catholic Brain Health Center, Yeouido St. Mary’s Hospital, and Eunpyeong St. Mary’s Hospital at the Catholic University of Korea. The dataset for brain segmentation is approved under IRB No. SC20RISI0198 as confirmed by Dr. Hyun Kook Lim on 31 December 2020. The dataset for WMH segmentation is approved under IRB No. PC20EISI0094 as confirmed by Dr. JeeYoung Kim on 2 July 2020.

For brain tissue segmentation, a total of 68 subjects with paired T1w MRIs and FLAIR MRIs were obtained from Yeouido St. Mary’s Hospital and Eunpyeong St. Mary’s Hospital from 2017 to 2019. The T1w MRIs have a consistent matrix size of 256 × 256 × 256 along with a pixel spacing of 1.0 × 1.0 × 1.0 mm${}^{3}$. Moreover, the paired FLAIR MRIs have a consistent matrix size of 348 × 384 × 28 along with a pixel spacing of 0.57 × 0.57 × 6 mm${}^{3}$.

For WMH segmentation, a total of 396 FLAIR MRIs with clinically confirmed WMH regions and annotated labels were initially obtained from Eunpyeong St. Mary’s Hospital from 2020 to 2021. The MRIs have a consistent matrix size of 768 × 768 × 32 and a pixel spacing of 0.27 × 0.27 × 5 mm${}^{3}$. However, because of the various levels of WMH in the dataset, we excluded the FLAIR MRIs with neglectable WMH regions along with mislabeled cases, obtaining a final total of 308 FLAIR MRIs.

### 2.2. Overview of the Proposed Method

Our goal is to produce brain tissue and WMH segmentation exclusively on FLAIR MRI. However, it is impractical to generate the ground truth using FLAIR MRI because it lacks structural information when compared with T1w MRI. Therefore, we suggest the following steps, as shown in Figure 1: (A) Brain Tissue Segmentation from FLAIR MRI: T1w MRI brain tissue labels are generated using FreeSurfer and the labels are co-registered with the FLAIR MRI. (B) Brain Tissue Segmentation Enhancement: the co-registered pseudo-labels are enhanced with an initial semi-supervised learning method using a deep learning segmentation architecture followed by a morphological correction. (C) Brain Tissue and WMH Segmentation: this process trains the brain tissue and WMH segmentation processes individually and merges the results into one label with their predictions.

### 2.3. Brain Tissue Segmentation from FLAIR MRI: Pseudo-Labeling-Based Segmentation

#### 2.3.1. Pseudo-Labeling from T1w MRI

For the T1w MRIs, we used FreeSurfer (6.0, Boston, MA, USA) with a “recon-all” pipeline, then extracted the brain labels as the pseudo-labels, which consist of cerebral gray matter, cerebral white matter, cerebellum gray matter, cerebellum white matter, and lateral ventricle from aseg+aparc.mgz [9].

#### 2.3.2. Co-Registration

Co-registration is a method that aligns two individual MRIs (e.g., different modalities) obtained from the same subject. In our case, this is used to align the T1w MRI with the FLAIR MRI. Because the primary purpose of the first step of our process is to generate initial brain tissue labels on the FLAIR MRI, we calculated the transform matrix from the T1w MRI to FLAIR MRI using a spatial co-registration method with rigid transformation from the SimpleITK library [14]. We transformed the pseudo-labels to the FLAIR MRI using the registered transform matrix. However, because of differences in the MRI spacings and dimensions, the result did not delineate brain tissue structure accurately. Therefore, we iteratively enhanced the brain tissue segmentation labels of the FLAIR MRIs.

### 2.4. Brain Tissue Segmentation Enhancement

#### 2.4.1. Deep Learning-Based Initial Segmentation

We trained a convolutional neural network (CNN) with the FLAIR MRI and co-registered pseudo-labels of the brain tissue as shown in Figure 2. For the initial segmentation model, we used U-Net [15] with an evolving-normalization (EvoNorm) activation layer [16]. In the preprocessing step, the histogram-based intensity regularization, min-max normalization with percentile cut-offs of (0.05, 99.95), and z-score normalization were performed, and the input shape was set as 196 × 196. We used medical MRI-based augmentation techniques to improve the robustness of the CNN-based segmentation architecture. Our data augmentations were applied using TorchIO [17].

#### 2.4.2. Morphological Label Correction

After training on the brain tissue segmentation in the FLAIR MRIs, there remains some noise that makes the training label data incomplete. Thus, we perform a simple morphological correction method based on brain structure characteristics to enhance the brain tissue labels. In addition, we performed connected component-based noise reduction using the fill-hole method [18] by connecting the nearest 26 voxels in three dimensions. The morphologically processed brain tissue label does not have either isolated labels or holes.

### 2.5. WMH Segmentation: Supervised Learning-Based Segmentation

#### 2.5.1. Annotated Labeling with Radiologists

Reference segmentation of the WMH was performed by manual outlining on the FLAIR MRIs. A total of 308 FLAIR MRI datasets were manually segmented, producing binary masks with a value of 0 (non-WMH class) or 1 (WMH class). The manual segmentation process was performed through the consensus of three certified radiologists (J.Y. Kim, S.W. Oh, and M.K. Lee) who did not have access to the T1w MRIs for the subjects. For the process of consensus, two radiologists discussed the criteria used for defining manual segmentation and had training sessions to standardize their visual skills. Manual segmentation was performed independently, and the segmentation results were then exchanged for confirmation. Chronic infarcts were hypointense with hyperintense rim lesions on the FLAIR MRI and were excluded from WMH labeling.

#### 2.5.2. Preprocessing

We performed the resampling into a 1 mm${}^{2}$ isometric space (z-direction was excluded from this process because our training process requires a 2D slice MRI). For WMH segmentation, we used the skull-stripping method with HD-BET [19] on the FLAIR MRIs of the WMH dataset to focus the regions corresponding to our training on the white-matter regions. Moreover, as shown in Figure 3, we used histogram-based intensity regularization, min-max normalization with percentile cut-offs of (0.05, 99.95), and z-normalization using TorchIO [17] to deal with the differences in MRI intensity variance.

### 2.6. Training

We compared three well-known segmentation architectures for the brain tissue segmentation and WMH segmentation: U-Net [15], U-Net++ [15], and HighRes3DNet [20]. We used the kernel sizes specified in the original article for each architecture. We set the input and output shapes to 196 × 196 and used the EvoNorm activation layer [16] instead of batch normalization and an activation function. The only difference between the training processes of brain tissue segmentation and WMH segmentation is the data augmentation.

For brain tissue segmentation, we used the following TorchIO augmentation methods [17].

RandomAffine, which has a scale parameter in the range of 0.85–1.15.RandomMotion, with a degree value up to 10 and a translation value up to 10 mm.RandomBiasField, with a magnitude coefficient parameter ranging between −0.5 and 0.5.RandomNoise, which has a mean value of Gaussian distribution in range of 0 to 0.025.RandomFlip, with a spatial transform value up to 2, which inverts the Z axis.

For WMH segmentation, we did not include data augmentation in our method so that we could focus on the actual intensity range of the WMH regions.

### 2.7. Experiment Setup

We used the PyTorch deep learning library [21] for our main framework on a workstation with an Intel i9-9900X 3.5 GHz CPU, 128G RAM, and two NVIDIA RTX 2080 11 GB GPUs. In addition, for preprocessing and augmentation, we used TorchIO library [17].

For brain tissue segmentation enhancement (Figure 1B), we trained the CNN-based segmentation model using patches that consist of 128 samples per FLAIR MRI cropped from randomly selected locations. The size of the cropped MRI patches is 128 × 128. In addition, we used the cross-entropy loss function [22] and AdamW optimizer [23] with learning rate = 0.001 and weight decay = 0.01.

For brain tissue segmentation, we divided 68 subjects with a split ratio of 0.8; 54 subjects were used for training and 14 subjects were used for validation. After model training was complete, we used grid-based sampling and aggregation to perform inference on the segmentation results.

For WMH segmentation, we distributed 308 subjects, using 277 subjects for training and 31 subjects for validation. No medical MRI-based augmentation technique was used to process the WMH segmentation to avoid confusing any of the information on the already sensitive object. The remainder of the experiment setup was the same as that of brain tissue segmentation. For the loss function, we used the DiceBCE loss function, which is a combination of the Dice loss function [24] and binary cross-entropy loss function [22], to handle the varying sizes of WMH regions. Moreover, we used the AdamW optimizer [23] with learning rate = 0.001 and weight decay = 0.01.

### 2.8. Metrics for Evaluation

#### 2.8.1. Evaluation for Brain Tissue Segmentation

The co-registered pseudo-labels of brain tissue cannot be interpreted as the ground truth due to noise, isolated regions, and misled hole results from using the co-registration method. Therefore, we measured the relative volume difference between the labels from T1w MRI using FreeSurfer and the labels predicted from FLAIR MRI using our proposed method. The relative volume difference is defined as follows:$$\mathrm{Relative}\;\mathrm{Difference}\;\left(X,X_{reference}\right)=\frac{|X-X_{reference}|}{X_{reference}}*100$$

#### 2.8.2. Evaluation for WMH Segmentation

To evaluate the performance of WMH segmentation, we measured the Dice overlap score [25], which measures the similarity between the ground truth label and the prediction label.
$$\mathrm{Dice}\;\mathrm{Overlap}\;\mathrm{Score}\;\left(A,B\right)=\frac{2\left|A\cap B\right|}{\left|A\right|+\left|B\right|}$$

Because of the variance in size in the WMH segmentation, we expect several false positives and false negatives in the predicted segmentation of WMH. Therefore, we also measured the ratio of true positives to positive predictions (the precision), the ratio of true positives to all predictions (the recall), and the weight of precision and recall (the F1 score).
$$\mathrm{Precision}=\frac{TruePositive}{TruePositive+FalsePositive}$$
$$\mathrm{Recall}=\frac{TruePositive}{TruePositive+FalseNegative}$$
$$F1\;\mathrm{Score}=2*\frac{Precision*Recall}{Precision+Recall}$$

## 3. Results

### 3.1. Measured Volume Comparisons: Brain Tissue Segmentation

Figure 4 presents the result of brain tissue segmentation from each model. By comparing each model’s predicted label with its pseudo-label, we can observe that all three models (U-Net++, U-Net, and HighRes3DNet) have made changes in initially empty regions.

By comparing the volume obtained by each model to the volume obtained by FreeSurfer from the T1w MRI, we can measure the average relative volume difference (U-Net++, 4.8 ± 2.0; U-Net, 4.8 ± 1.7). HighRes3DNet tends to underestimate Cerebellum GM and overestimate Cerebellum WM (Cerebellum GM volume: Reference, 430.8 ± 45.7; HighRes3DNet, 408.0 ± 34.6. Cerebellum WM volume: Reference, 499.3 ± 5.8; HighRes3DNet, 559.3 ± 56.4). The average relative volume difference between the results of HighRes3DNet and FreeSurfer is 7.4 ± 3.4. Other detailed volumes and relative volume differences are listed in Table 2 and Table 3.

### 3.2. Dice Overlap Scores: WMH Segmentation

Figure 5 shows the results of WMH segmentation from each model. For comparably large WMH regions, all three models had predictions similar to the ground truth. However, many false positives and false negatives were found in small WMH regions.

As shown in Table 4, U-Net performed best on the dice overlap score with 0.81 ± 0.07, and f1 score with 0.84 ± 0.04. HighRes3DNet performed the best on recall with 0.92 ± 0.06, yet the score showed the lowest dice overlap score. As Figure 6 demonstrates, U-Net had the best performance with the lowest range of the interquartile range (IQR) and its whiskers.

## 4. Discussion

In this study, we developed a reliable automated segmentation method using FLAIR for WMH and cortical volume without the need for 3D T1w MRIs. In clinical practice, it can be difficult to obtain 3D T1w MRIs because of the long scan time, magnetic resonance machine performance, and the patient’s condition. In contrast, FLAIR MRI is a more common and essential sequence for evaluating the brain and is easy to obtain in routine practice, so this method is applicable to more patients.

As it is shown in Figure 7, the result of brain tissue segmentation enhancement suggests that semi-supervised learning was able to process the direct training of brain tissue segmentation only on FLAIR MRI. By following the procedure shown in Figure 1, the transformation between two different MRI modalities (T1w and FLAIR) could be made with improvements to the labeling quality. As we intended, our method was able to measure the volume of brain tissue and WMH using only FLAIR MRI.

### 4.1. Performance of Brain Tissue Segmentation

We evaluated the performance of the segmentation by comparing the relative difference in volume of the results obtained using T1w MRI and its paired FLAIR MRI. However, the difference in relative volumes was less than 10% for all three models: 0.86 for U-Net, 0.85 for U-Net++, and 0.81 for HighRes3DNet. Considering that a relative difference of 3.4% already existed in the pseudo-labels for FLAIR MRI, we conclude that our method is sufficient for FLAIR MRI segmentation.

However, we noticed that our segmentation method has a limitation due to the absence of ground truth. Therefore, in a further study, we would like to evaluate our generated brain tissue pseudo-labels using radiologists and measuring the Dice overlap score.

### 4.2. Performance of WMH Segmentation

We compared the Dice overlap between the ground truth and the prediction for each model. We figured U-Net, with a Dice overlap score of 0.81 ± 0.07, a precision of 0.88 ± 0.05, a recall of 0.80 ± 0.08, and an F1 score of 0.83 ± 0.05, is the most balanced segmentation architecture of the comparison models.

Even though U-Net++ and HighRes3DNet had lower dice overlap scores compared to U-Net, as shown in Figure 6, we could still demonstrate that any convolutional network-based segmentation architecture is qualified for WMH segmentation.

### 4.3. Clinical Relevance and Application

In clinical practice, the proposed algorithm will be a useful screening tool for the quantification of cortical volume and WMH burden in the elderly with cognitive impairment using only 2D FLAIR MRIs and without the need for 3D T1 volume MRIs. This will be beneficial because it is difficult to obtain 3D T1 volume MRIs in the elderly because of the long scan time and need for a high-performance magnetic resonance machine to obtain good quality data.

T2, SWI, and GRE are also known for the difficulty of obtaining brain tissue labels from these modalities. This is because it is effectively impossible to obtain structural information from these MRIs without the paired T1w MRI. As mentioned before, there are many medical segmentation studies based on deep learning, but few considered obtaining the structural information from any single modality other than T1w MRI.

## 5. Conclusions

We introduced a semi-supervised learning method for brain tissue segmentation using only FLAIR MRI. With our brain segmentation results, we demonstrated that our FLAIR MRI segmentation is just as reliable as segmentation using its paired T1w MRI. We moreover showed that brain tissue segmentation and WMH segmentation could be performed from a single FLAIR MRI. Furthermore, the results indicate that our semi-supervised learning method is not limited to FLAIR MRI but could also be applied to T2, SWI, and GRE MRIs without the need to obtain brain tissue labels from paired T1w MRIs. We believe our semi-supervised learning method has the clinical potential of being a key solution for quantifying cortical volume and WMH burden for WMH analysis using FLAIR MRI exclusively.

## Figures and Tables

**Figure 1 brainsci-11-00720-f001:**
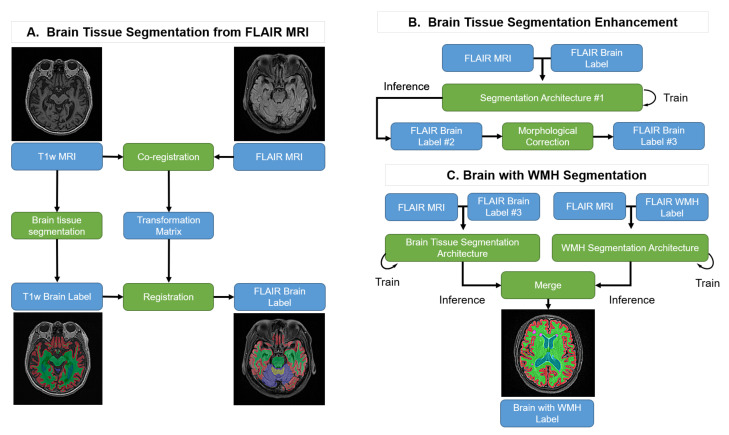
Pipeline of the proposed method. Blue boxes: input or output data; Green boxes: computational processes.

**Figure 2 brainsci-11-00720-f002:**
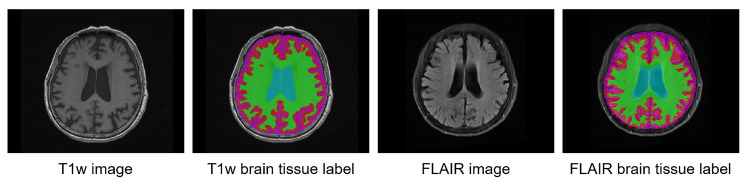
Example of brain tissue labels on T1w MRI and their co-registered pseudo-labels on FLAIR MRI.

**Figure 3 brainsci-11-00720-f003:**
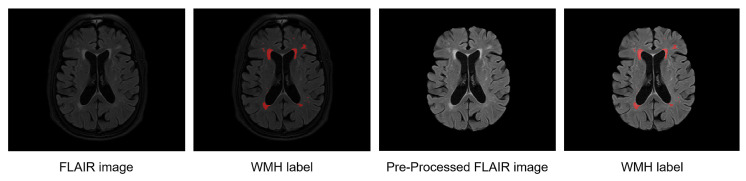
Example of WMH labels on a raw FLAIR MRI and WMH labels on the preprocessed FLAIR MRI.

**Figure 4 brainsci-11-00720-f004:**
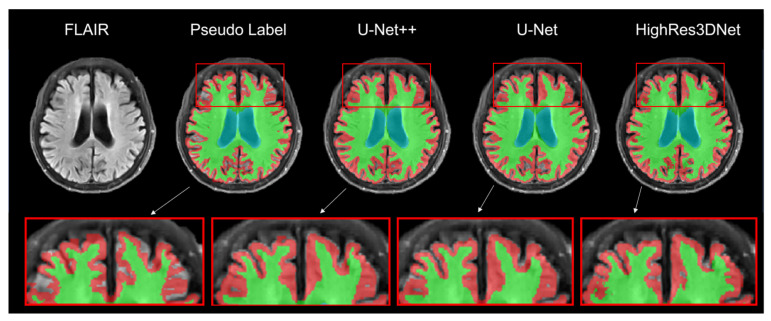
Result of brain tissue segmentation of the same subject in the axial view. The red, green, and blue labels represent the gray matter, white matter, and ventricle regions. Moreover, the highlighted boxes show the differences in the segmentation labels for each model.

**Figure 5 brainsci-11-00720-f005:**
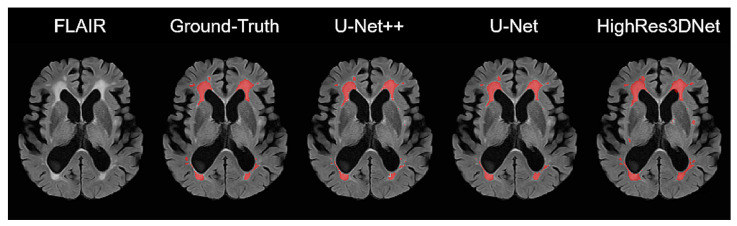
Comparison of the WMH reference and the predicted segmentation from each model (U-Net++, U-Net, and HighRes3DNet).

**Figure 6 brainsci-11-00720-f006:**
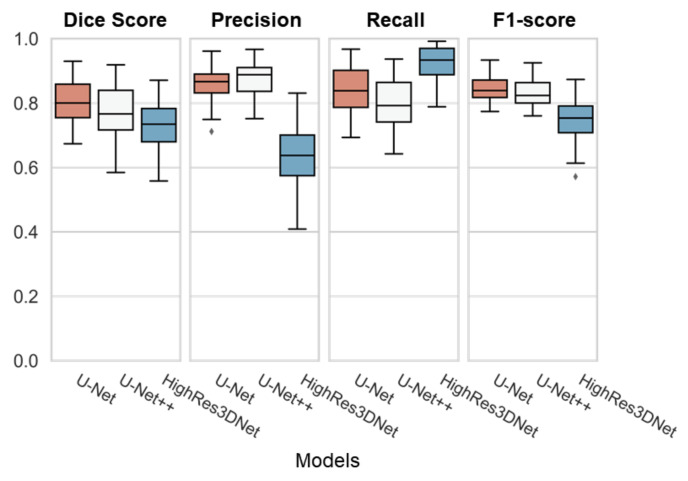
Boxplot of all three models (U-Net, U-Net++, and HighRes3DNet) with individual evaluations (Dice Score, Precision, Recall, and F1-score). Each box represents the interquartile range (IQR) with its median line in the box. The whiskers represent the range of extended IQR by 1.5 times, and the dot outside of the whiskers represents the outlier.

**Figure 7 brainsci-11-00720-f007:**
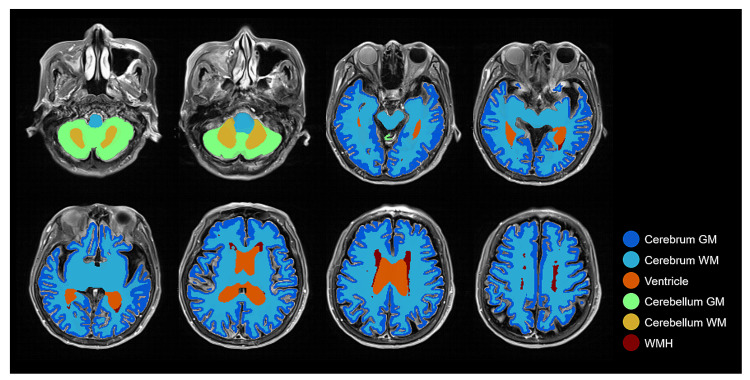
Combined brain tissue and WMH label on FLAIR MRI with several layers demonstrating all labels: Cerebrum GM, Cerebrum WM, Ventricle, Cerebellum GM, Cerebellum WM, and WMH.

**Table 1 brainsci-11-00720-t001:** Summary of the datasets used for training and validation.

Dataset	MRI	No. of Subjects	Matrix Size	Pixel Spacing (mm)	Purpose
CABI	T1w	68	256 × 256 × 256	1.0 × 1.0 × 1.0	Brain tissue segmentation
CABI	FLAIR	68	348 × 384 × 28	0.57 × 0.57 × 6	Brain tissue segmentation
CABI	FLAIR	308	768 × 768 × 32	0.27 × 0.27 × 5	WMH segmentation

**Table 2 brainsci-11-00720-t002:** Volume of individual brain tissue from the FreeSurfer labels (T1w MRI), pseudo-labels, U-Net++, U-Net, and HighRes3DNet.

Measurement	Brain Tissue	FreeSurfer Label (T1w)	Pseudo Label (FLAIR)	U-Net++	U-Net	HighRes3DNet
Volume (mL, mean ± SD)	Cerebellum GM	430.8 ± 45.7	444.5 ± 47.3	458.0 ± 45.4	455.4 ± 42.0	408.0 ± 34.6
	Cerebellum WM	499.3 ± 55.8	516.3 ± 57.5	510.4 ± 52.7	519.5 ± 54.4	559.3 ± 56.4
	Cerebrum GM	100.2 ± 10.2	103.5 ± 10.7	102.2 ± 9.6	101.1 ± 9.6	96.7 ± 8.9
	Cerebrum WM	23.2 ± 3.0	24.0 ± 3.1	22.5 ± 3.1	23.7 ± 2.8	22.9 ± 3.6
	Lateral Ventricles	41.2 ± 20.8	42.5 ± 21.6	39.1 ± 20.6	39.9 ± 20.7	41.5 ± 21.3

**Table 3 brainsci-11-00720-t003:** Relative difference between the FreeSurfer labels (T1w) and in the pseudo-labels, U-Net++, U-Net, and HighRes3DNet. GM, gray matter; SD, standard deviation; WM, white matter.

Measurement	Brain Tissue	Pseudo Label (FLAIR)	U-Net++	U-Net	HighRes3DNet
Relative Difference (%, mean ± SD)	Cerebellum GM	3.2 ± 1.1	6.4 ± 2.5	5.9 ± 2.8	5.4 ± 4.0
	Cerebellum WM	3.4 ± 0.9	2.5 ± 2.0	4.1 ± 2.0	12.2 ± 3.6
	Cerebrum GM	3.3 ± 1.4	2.7 ± 2.2	2.3 ± 1.7	4.3 ± 3.1
	Cerebrum WM	4.3 ± 3.2	6.1 ± 3.7	6.8 ± 6.7	9.7 ± 7.9
	Lateral Ventricles	3.1 ± 1.4	6.1 ± 4.3	4.7 ± 4.3	5.4 ± 6.0
Average Difference (%, mean ± SD)	-	3.4 ± 0.5	4.8 ± 2.0	4.8 ± 1.7	7.4 ± 3.4

**Table 4 brainsci-11-00720-t004:** Average dice overlap score of WMH, along with precision, recall, and F1 score value.

	Dice Overlap Score	Precision	Recall	F1 Score
U-Net++	0.77 ± 0.09	0.88 ± 0.05	0.80 ± 0.08	0.83 ± 0.05
U-Net	0.81 ± 0.07	0.86 ± 0.06	0.84 ± 0.08	0.84 ± 0.04
HighRes3DNet	0.73 ± 0.07	0.64 ± 0.09	0.92 ± 0.06	0.75 ± 0.07

## Data Availability

Not applicable.

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
