# Peer review of "Semi-Supervised Learning in Medical MRI Segmentation: Brain Tissue with White Matter Hyperintensity Segmentation Using FLAIR MRI"

_brainsci, 2021, doi:10.3390/brainsci11060720_

Round 1

Reviewer 1 Report

The paper focused on a method, based on semi-supervised learning, aiming to achieve a volumetric measurement of white matter hyper intensity on FLAIR images. The method described is interesting and may have a relevant clinical impact however the paper needs some structural corrections to enhance its readability.

 The introduction contains a misleading information about the availability of 3D FLAIR sequences in clinical practice. The use of 3D FLAIR sequences in Neurooncology is well documented in literature as well as in clinical reality. The paragraphs on brain segmentation on T1 rather than on other MR sequences should be formally and conceptually edited and the references implemented accordingly. Volumetric segmentation of FLAIR hyperintense areas is a reality in neurosurgery and radiotherapy; and this should be mentioned in introduction.

Methods are detailed and each step described in separated paragraph. However a clearer explanation of labeling process, segmentation and learning should be provided, in order to enhance the informative value of the paper and to make the proposed method understandable and reproducible. 

Discussion must include a section highlighting the difference between the proposed method and the manual segmentation of FLAIR hyperintensity already reported and used in Neuro-oncology. Moreover, the clinical importance of the proposed method should be better supported by the literature and some speculations on future application should be included.

Also conclusion section should contain information on clinical potentiality of the proposed method.

Reviewer 2 Report

The manuscript describes a method to segment FLAIR images and identify WMH in absence of T1 images collected. It is well-written and is potentially informative for the field. I have several general comments about this manuscript:

Major:

  1. I hope the authors can more clearly highlight the novelty of this study. I understand the author’s approach segments WMH but also obtains labels on FLAIR images. But can authors elaborate on why knowing the label on the FLAIR is important to understand WMH. And by labeling do authors mean labeling just the grey/white matter/CSF?
  2. Have the authors compared their approach to existing approaches automatically quantifying the WMH, e.g. https://www.biorxiv.org/content/10.1101/2020.10.17.343574v2.full. How did the authors decide on using U-Net? Have the authors compared to other segmentation models such as Segnet and Mask R-CNN? I am also wondering what the exact model Freesurfer use to segment the T1 images. Can that same model be adapted to work on FLAIR for consistency? For the 68 patients that have both T1 and FLAIR, can the authors compare the results using Freesurfer in identifying the WMH (hypo-intensity in this case) on T1 to using their method on FLAIR?
  3. Is the code for processing, data, or the trained model shared publicly? Without that, it is difficult for readers to evaluate the models based solely on the descriptions in the manuscript and the segmentation protocol cannot be used by other researchers.
  4. The method section is written with many technical details. I recommend the authors to give more space to general descriptions of rationale and steps first and then go into specifics. I also recommend reducing technical details. Since not all readers are familiar with the specific models, it is best to state what authors did use than what they did not use (e.g. P5 Line 133). Sharing code could also help reducing confusion in these details.
  5. Relatedly, I understand the morphological label correction is necessary due to some mislabeling. But how did the authors quantify that after the correction the results are “better”?
  6. As I understand, the segmentation boundaries can vary a lot by the person conducting the segmentation. How did the radiologists achieve “consensus”? Do they all segment the same image and is there a reliability value (ICC) that can be reported?
  7. Why for MH segmentation only one model was used instead of all 3?
  8. Please briefly explain why this study is a “semi-supervised” learning.
  9. Page 4 Line 99, is the “T1w brain segmentation tool” just Freeserfer? Please elaborate.
  10. The author speculated if given more time HighRes3DNet can have similar results as U-Net, but this is a testable hypothesis.

Minor

There are some language issues throughout. I note some of them:

  1. Page 2 Line 46, the “Burden” should not be capitalized.
  2. Page 5 Line 116, the “rather”.
  3. Page 6 Line 162, the “location4”.
  4. “Mislabeled” is spelled “miss-labeled” in a few places.

Round 2

Reviewer 2 Report

It is helpful to highlight changes in the manuscript between reviews.

I have no further comments.